# Novel Extract from Beetle *Ulomoides dermestoides*: A Study of Composition and Antioxidant Activity

**DOI:** 10.3390/antiox10071055

**Published:** 2021-06-30

**Authors:** Nina A. Ushakova, Efim S. Brodsky, Olga V. Tikhonova, Alexander E. Dontsov, Maria V. Marsova, Andrey A. Shelepchikov, Alexander I. Bastrakov

**Affiliations:** 1A.N. Severtsov Institute of Ecology and Evolution RAS, Leninsky Prospect, 33, 119071 Moscow, Russia; ushakova@sevin.ru (N.A.U.); efbr@mail.ru (E.S.B.); dioxin@mail.ru (A.A.S.); aibastr@yandex.ru (A.I.B.); 2Institute of Biomedical Chemistry (IBMC), Pogodinskaya Str., 10, 119121 Moscow, Russia; ovt@ibmh.msk.su; 3N.M. Emanuel Institute of Biochemical Physics, RAS, Kosygina Str., 4, 119334 Moscow, Russia; 4N.I. Vavilov Institute of General Genetics, Russian Academy of Sciences, Gubkina Str., 3, 119991 Moscow, Russia; masha_marsova@mail.ru

**Keywords:** *Ulomoides dermestoides*, electro-pulse plasma dynamic extraction, antioxidant activity, lifespan, antiglycation activity

## Abstract

A biologically active extract from the darkling beetle *Ulomoides dermestoides* was obtained using the electro-pulse plasma dynamic extraction method. The beetle water extract contained a complex of antioxidant substances such as antioxidant enzymes and nonprotein antioxidants, as well as a complex of heat shock antistress proteins. This determines the rather high antioxidant activity of the aqueous extract of the beetle, i.e., 1 mg of dry matter/mL of the extract has an equivalent antioxidant activity to 0.2 mM Trolox (a water-soluble analog of vitamin E). It was shown that the beetle extract can lead to a 25–30% increase in the average lifespan of nematode *Caenorhabditis*
*elegans*, under normal conditions, and a 12–17% increase under conditions of oxidative stress (with paraquat), and significantly inhibits the fructosylation reaction of serum albumin. Therefore, the beetle aqueous extract shows promise as a biologically active complex exhibiting antioxidant activity.

## 1. Introduction

Insects that show promise for the development of new antibiotics, immunoactive peptides, and substances for pharmaceutical use have been attracting the attention of researchers. Interest in the use of the biomass of certain insect species for the production of biologically active substances has been enhanced by the development of technologies for breeding some species in artificial conditions. In this case, it is important to identify the active components of insects and obtain evidence of their biological effects on the body.

Insects are a natural source of substances with biological activity, including antimicrobial, antiviral, anti-inflammatory, antioxidant, and anticancer effects [1]. In Central and South American traditional medicine, the darkling beetle *Ulomoides dermestoides* is used to treat inflammatory diseases and cancer. The literature has also reported on the anti-inflammatory and immunomodulatory activities of an aqueous extract of this beetle, suggesting the presence of components that are pharmacologically promising [2,3,4,5,6,7,8]. The insect is able to reproduce rapidly in artificial conditions similar to those described for the related species *Alphitobius diaperinus* [9]. This indicates the possibility of obtaining renewable biomass of the darkling beetle. However, in order to assess the possibility of its practical use, it is necessary to obtain characteristics of its composition and assess its biological activity. To maximize the extraction of useful components from biomass, it is important to use efficient extraction processes. Electro-pulse plasma dynamic extraction (EPDE) is a known extraction method that resembles a natural lightning storm, in which a high-voltage discharge is used to destroy biological objects [10]. When a high-power electric discharge is passed, shock waves appear in the extracted mixture, creating high impulsive pressure and powerful cavitation processes. Under the action of the forces caused by a high-voltage discharge (38,000 volts) in the extracted mixture (biological raw material and extractant), shock waves are generated within milliseconds, creating a high impulse pressure. As a result, in a millisecond time interval in the confined space of the reactor, a large amount of energy is released, destroying the cellular structures—external and internal biological membranes.

It is important to note that the extraction process using the EPDE method, in contrast to the traditionally used methods, takes place without heating. This is due to the fact that when an electric discharge passes through the extracted mixture, a gas–vapor film is formed from the evaporating liquid, which protects the organic raw material from combustion. Thus, biological raw materials are not subject to thermal damage, keeping all the extracted biologically active substances in their native state. Tissue processing in this way increases the yield of cellular components into the extract while maintaining biological properties. The EPDE method made it possible to obtain an extract with antioxidant activity from the larvae of the black soldier fly *Hermetia illucens* [11].

To study the biological activity of the aqueous extract, a complex approach was used, including analyses of the proteins and nonprotein hydrophilic components of the extract, followed by an assessment of its antioxidant activity in model systems. An indispensable tool for the characterization of proteins in biological samples is liquid chromatography coupled with the tandem mass spectrometry (LC–MS/MS) technique [12,13]. A proteomic analysis approach called GeLC–MS/MS was chosen to study the aqueous extract of *U. dermestoides* [14,15,16]. The nonprotein components of the extract were investigated using ultra-high-resolution chromatography–mass spectrometry.

The aim of this work was to comprehensively study the composition and biological activity of an aqueous extract from the biomass of adult sexually mature *U. dermestoides* individuals.

The antioxidant activity of the extract was evaluated in various model experiments, including quenching the chemiluminescence of luminol and, in particular, comparison with the action of the vitamin E analog Trolox. Testing for antioxidant effects was carried out in the nematode *Caenorhabditis elegans* under conditions of oxidative stress induced by paraquat toxin. The study of nonenzymatic antiglycation activity may indicate a connection with the presence of antioxidants in the extract, while also indicating possible therapeutic and prophylactic effects.

## 2. Materials and Methods

### 2.1. Reagents

All chemicals used in these experiments were of analytical grade and were commercially available. All solvents used were HPLC grade, purchased from Sigma-Aldrich (Chemical Co., St Louis, MO, USA), and Merck Life Science, Darmstadt, Germany. Reagents were purchased from Sigma-Aldrich and Merck. Plastic disposable tubes and pipettes were purchased from Eppendorf Corporate, Hamburg, Germany.

### 2.2. Materials

#### 2.2.1. Obtaining Water Extracts of the Beetle

The darkling beetle *U. dermestoides* was raised on a dry nutritional mixture of wheat bran (70%) with dry milk (5%), cornflour (20%), and sunflower meal (5%) in a climatic chamber at a temperature of 28 °C and a humidity of 60–70%. Adult sexually mature beetles of both sexes were immobilized by cooling at −18 °C, then crushed (Tube Mill 100 control, Labicon, Samara, Russia) and extracted by the EPDE method in distilled water at 23 °C; EPDE extraction was carried out on a setup from KorolevPharm LLC, Russia. Extraction parameters were as follows: the power of the transmitted electric discharge, 38,000 volts; the pulse frequency, 1 pulse/s; distance between the electrodes, 5 mm; extraction time, 7 min. After exposure, the extract was separated from the solid fraction by centrifugation at 5000× *g* for 15 min. Antibacterial treatment was carried out by membrane filtration using Millex-GV 0.22 μm filtering attachments. The extract was stored at −18 °C.

#### 2.2.2. Cultivation of Nematodes

*C. elegans* nematodes, strain Bristol N2 (wild type), provided by Elena Budovskaya (University of Amsterdam, Amsterdam, Netherlands), were cultured on solid agar NGM medium at 20 °C according to the method described by T. Stiernagle [17]. A synchronized nematode population was obtained using the method of Solis and Petrascheck [18]. All manipulations with nematodes were carried out according to standard techniques [19]. The model system was adapted for the analysis of the investigated substances in a liquid medium [20]. The standard *E. coli* strain op50 for feeding nematodes was grown at 37 °C under aerobic conditions on solid agar medium NGM or in Luria–Bertani broth (LB), medium pH (7.5 ± 0.2; at 25 °C) for 18–20 h. Paraquat dichloride, SIGMA (methyl viologen dichloride hydrate) was used as an inducer of oxidative stress [21].

The study was conducted according to the guidelines of the Declaration of Helsinki and approved by the Ethics Committee of A.N. Severtsov Institute of Ecology and Evolution of the Russian Academy of Sciences (Protocol Code: 009; Date of Approval: 26 October 2020).

### 2.3. Proteomic Analysis of an Aqueous Extract of the U. dermestoides Beetle

#### 2.3.1. SDS–PADE

Before the experiment, the sample was thawed for 30 min in a water bath at a temperature of +25 °C. Denaturing one-dimensional electrophoresis was carried out in the presence of SDS with a polyacrylamide concentration of 8 to 12% in a separating gel and 5% in a concentrating gel. A total of 8 μL of the sample was applied to the lane, and electrophoresis was carried out for 60 min at constant current (18 mA). In this case, the voltage was 50 V for the first 10 min and increased to 200 V from 11 to 60 min. The samples were fixed with a solution of acetic acid and ethanol (2 times for 20 min) and then stained with Coomassie’s solution (1 h) and washed overnight. The gels were scanned using an ImageScanner III (GE Healthcare Bio-Sciences Corp., Piscataway, NJ., USA). The protein line was cut into six approximately equal fragments about 2 mm wide and about 10–12 mm long. The obtained gel fragments were washed 2 times for 15 min with 100 μL of 50 mM ammonium bicarbonate solution, then dried with 100% acetonitrile (15 min), after which the solution was removed, and the samples were dried on a centrifugal vacuum evaporator (Concentrator Plus, Eppendorf). Proteins in each gel fragment were distained and digested by trypsin using the protocol described previously [22].

#### 2.3.2. LC–MS/MS

The LC–MS/MS of obtained peptides was performed through the method described below. A volume of 1 µL peptides was loaded onto the Acclaim µ-Precolumn (0.5 mm × 3 mm, 5 µm particle size, Thermo Scientific, Rockwell, IL, USA) at a flow rate of 10 µL/min for 2 min in an isocratic mode of Mobile Phase C (2% acetonitrile, 0.1% formic acid). Then, the peptides were separated with high-performance liquid chromatography (HPLC, Ultimate 3000 Nano LC System, Thermo Scientific, Rockwell, IL, USA) in a 15 cm long C18 column (Acclaim^®^ PepMap™ RSLC inner diameter of 75 μm, Thermo Fisher Scientific). The peptides were eluted with a gradient of buffer B (80% acetonitrile, 0.1% formic acid) at a flow rate of 0.3 μL/min. The total run time was 60 min, which included an initial 2 min of column equilibration to 2% of buffer B, then a gradient from 2 to 35% of buffer B over 43 min, then 3 min to reach 99% of buffer B, flushing 4 min with 99% of buffer B, and then 3 min to reach initial conditions (2% of buffer B) and 5 min re-equilibration.

MS analysis was performed in triplicate with a Q Exactive HF-X mass spectrometer (Q Exactive HF-X Hybrid Quadrupole-OrbitrapTM Mass spectrometer, Thermo Fisher Scientific, Rockwell, IL, USA). The temperature of the capillary was 240 °C, and the voltage at the emitter was 2.1 kV. Mass spectra were acquired at a resolution of 120,000 (MS) in a range of 300−1500 *m/z*. Tandem mass spectra of fragments were acquired at a resolution of 15,000 (MS/MS) in the range from 100 *m/z* to *m/z* value determined by a charge state of the precursor but no more than 2000 *m/z*. The maximum integration times were 50 msec and 110 msec for precursor and fragment ions, respectively. AGC target for precursor and fragment ions were set to 1 × 10^6^ and 2 × 10^5^, respectively. An isolation intensity threshold of 50,000 counts was determined for precursor selection and the top 20 precursors were chosen for fragmentation with high-energy collisional dissociation (HCD) at 29 NCE. Precursors with a charge state of 1+ and more than 5+ were rejected, and all measured precursors were dynamically excluded from triggering a subsequent MS/MS for 20 s.

The obtained raw data were processed using the MaxQuant software (version 1.6.3.4) with the Andromeda search engine [23]. The Uniprot database *Tribolium castaneum* was used to identify proteins of *U. dermestoides* by homology.

Protein sequences of the *Oryzias latipes* proteome provided by Uniprot (February 2021) were used for protein identification with the Andromeda engine. The identification settings were as follows: trypsin, as a specific protease with a maximum of 2 missed cleavage and a maximum *m/z* deviation of 5 ppm, was allowed for precursor identification, 10 ppm was set as the match tolerance for fragment identification (acquisition in Orbitrap); oxidation of methionines, N-terminal protein acetylation and modification of cysteine with propionamide were set as variable modifications for the peptide search. Peptide Spectrum Matches (PSMs), peptides, and proteins were validated at a 1.0% false discovery rate (FDR), estimated using the decoy hit distribution. Proteins were considered to be significantly identified if at least two peptides were found for them. Label-free protein quantification was based on iBAQ: the percentage of a specific protein was calculated as the ratio of the iBAQ value of the protein to the sum of all iBAQ values for all identified proteins, multiplied by 100%.

### 2.4. Analysis of the Nonprotein Portion of the Aqueous Extract of the Beetle

To study the nonprotein portion of the *U. dermestoides* extract, 50 μL of the extract was placed into an autosampler vial, 10 μL of a methanol solution of the internal standard D8-toluene (23.5 ng), and 1 mL of dimethyl ether were added and the mixture was extracted directly into the vial with vigorous shaking for 20 min. Then, 1 μL of the extract was injected into the gas chromatograph injector. The analysis was carried out using Q Exactive Orbitrap Mass Spectrometer (Thermo Fisher Scientific) with an SGE HT-8 silica capillary column (length 28 m, inner diameter 0.25 cm, a film thickness of the stationary phase 0.25 μm), temperature program 50 °C (2) −7 °C/min−295 °C (23 min), split-less (with purging after 0.05 min), scanning the mass spectrum in the range 40–500 at a resolution of 60,000. The injector temperature 260 °C, the carrier gas was He (0.8 mL/min). A total of 1 μL was injected from a volume of ~1 mL.

Peaks in chromatograms were identified by mass spectra using library search and expert judgment spectra–structure correlations and chromatographic retention indices [24]. Quantification was carried out on the basis of chromatographic peaks in the total ion current (TIC) mass chromatogram.

### 2.5. Determination of the Antioxidant Activity (AOA) of the Aqueous Extract of the Beetle

The AOA was determined by the usual method of quenching the chemiluminescence of luminol [25]. The maximum amplitude of chemiluminescence in the control and in the presence of different amounts of the extract was used as a measured parameter. The results were compared with the action of Trolox (6-hydroxy-2,5,7,8-tetramethylchromane-2-carboxylic acid, Fluka Chemie AG, Buchs, Switzerland), a water-soluble vitamin E analog, by measuring the dependence of the amplitude of chemiluminescence on the concentration of Trolox under the same conditions.

It was assumed that antioxidant enzymes—in particular, catalase—were present in the extract. Therefore, the effect of the original beetle extract (sample EXT) and the protein-free supernatant of the extract obtained by filtration on a filter that does not allow the passage of molecules with a molecular weight exceeding 10 kDa (sample SUP) were compared. For this, the original sample of the extract was diluted two times with K-phosphate buffer, and part of it was centrifuged in a Corning Spin-X UF tube at 15,000× *g* for 25 min. The resulting supernatant was brought up to the same volume that was used in the separation process. The concentrate containing high molecular weight substances, which did not pass through the filter, was washed from the filter and also brought to the original volume using the buffer (sample SED). It should be noted that the washing process might not have been complete, and some of the extract might have remained on the filter. The dry weight of the preparations was as follows: initial extract (EXT sample), 20.0 mg/mL; filter washout (SED sample), 16.7 mg/mL; supernatant (SUP sample), less than 1 mg/mL. To compare the antioxidant activity of these drugs, the kinetics of chemiluminescence quenching was measured at various volumes of added drugs. The reaction medium contained 0.05 M K-phosphate buffer (pH 7.4), 2.0 μM hemoglobin, 100 μM luminol, 100 μM EDTA, and various amounts of beetle extract. The reaction was started by adding 100 μM hydrogen peroxide. As a control, a buffer solution was used instead of the extract.

### 2.6. Measurement of Catalase Activity

The catalase activity of the extract was determined by measuring the content of hydrogen peroxide using ammonium molybdate [26]. The reaction medium contained 2.0 mL of 0.1 M K-phosphate buffer, 10–20 μL of beetle extract, and 12 mM hydrogen peroxide. The reaction was carried out for 1–3 min, after which it was stopped by the addition of 30 mM ammonium molybdate, and the optical density was measured at 410 nm. The control samples contained K-phosphate buffer instead of beetle extract. The concentration of hydrogen peroxide was determined from the calibration curve of the dependence of the optical density at 410 nm on the concentration of hydrogen peroxide. The catalase activity was expressed as the amount (μmol) of hydrogen peroxide decomposed in 1 min of 1 mg of the dry weight of the beetle extract at 20 °C. On average (according to the results of five measurements), the catalase activity of the beetle extract was 23.4 ± 3.5 μmol H_2_O_2_/min mg for the dry residue of the extract.

### 2.7. Determination of the Antiglycation Activity of the Aqueous Extract of the Beetle

Bovine serum albumin (BSA) was used as a glycation substrate. To modify BSA during glycation, fructose was used as a reducing sugar. Fructose is 8–10 times more reactive than glucose in the formation of Maillard reaction products [27,28]. Therefore, glycation in the presence of fructose (fructosylation), in contrast to glucose, leads to the accumulation of fluorescent protein adducts much faster. The incubation medium contained 0.1 M sterile potassium phosphate buffer (pH 7.4), 50 mM fructose, 4 mg/mL BSA, 2 mM sodium azide and beetle extract at various concentrations. As control samples, we used samples that did not contain beetle extract and samples that did not contain fructose. The samples were incubated at 37 °C in the dark with constant stirring for 3 days. After incubation, aliquots of samples were dialyzed against phosphate buffer in order to remove unreacted low molecular weight molecules. For dialysis, we used a Float-A-Lyser cellulose ether membrane (SPECTRUM Labs, United States), which allows molecules with a molecular weight of less than 3.5 kDa to pass through. Dialysis was carried out for 25 h at 6 °C. After dialysis, the intensity of the emission maximum fluorescence of the modified albumin was measured at a wavelength of 435 nm (excitation wavelength 365 nm).

### 2.8. Testing the Antioxidant Effect of the U. dermestoides Extract on the Nematode C. elegans under Paraquat-Induced Oxidative Stress

Synchronized sterile *C. elegans* eggs, obtained according to the standard method, were transferred to the bacterial lawn of the *E. coli* strain op50 and incubated for 3 days at 20 °C. The larvae of the L4 stage obtained in this way were carefully transferred to the buffer solution. Aliquots of S-medium, 8–12 nematodes (total number of nematodes per point, 60–90 specimens), an aliquot of the investigated aqueous extract, and an aqueous solution of the oxidative stress inducer (paraquat) were added to each well of a 96-well plate. The study of each extract was carried out under paraquat concentrations of 0 and 50 mM. We used the tested aqueous extract in three concentrations: not diluted, and diluted with a solution of S-medium by 4 or 10 times. The amount of extract added to the cell, in all variants, was 1/10 of the total volume of the experimental medium.

The antioxidant activity of the extract was assessed by the change in the median lifespan (MLS) of the nematode [29,30], that is, the time when the number of living individuals reached 50% of their initial number. Observations and counting of active nematodes were carried out every three hours until 80% of their initial number died. The number of live and dead nematodes was calculated based on their activity when exposed to light and touching a probe. MLS of nematodes was determined graphically when the graph crossed the value 0.5—half of the initial number of individuals participating in the experiment. The number of live nematodes introduced into the well was taken as 1. The MLS of the experimental group was compared with the MLS of the control group of nematodes grown under standard conditions or under conditions of paraquat-induced oxidative stress (OS).

Statistical analysis was performed using the Statistica 12 software. Using Statistica 12 software processed data on MLS of the nematode *C. elegans*.

## 3. Results

The aqueous extract of the *U. dermestoides* beetle contains various antioxidants, both proteins and nonprotein substances, as well as a complex of heat shock antistress proteins.

The first stage in the proteomic analysis of an aqueous extract of the *U. dermestoides* beetle was the electrophoretic separation in a polyacrylamide gel. Figure 1 shows the result of the SDS–PAGE protein separation and cutting scheme for six gel fragments.

Peptides obtained after in-gel digestion with trypsin were further analyzed by HPLC–MS/MS in at least two technical replicates for each gel fragment. SDS–PAGE, followed by HPLC–MS/MS, yielded 329 proteins that were significantly identified by homology in the extract. The following proteins were most represented in the extract of the *U. dermestoides* beetle: myosin heavy chain tr|A0A139WE70|A0A139WE70_TRICA, myosin heavy chain muscle-like protein (14.3%), actin-like protein tr|D6WF19|D6WF19_ETRICA actin-87 protein (7.9%), and beta subunit of ATP synthase tr|D2A4R3|D2A4R3_TRICA ATP synthase subunit beta (5.8%) of all identified proteins. Table 1 lists some proteins identified by homology. The antioxidant content in the extract was 3.43% in total, while heat shock proteins accounted for 1.3%.

The composition of the nonprotein components of the extract is presented in Table 2. It can be seen that the water extract contained antioxidants; the dominant component was ethyl p-hydroquinone, along with phenolic compounds (e.g., 3-methoxy-2-methylphenol and 3-methoxyphenol).

The presence of antioxidants and antistress proteins in the extract suggests its antioxidant activity, which was verified in model experiments on chemiluminescence quenching of luminol as well as in an experiment on the lifespan of the free-living nematode *C. elegans* under oxidative stress.

Figure 2A shows the comparative kinetics of the development and quenching of luminol chemiluminescence in the presence of the SUP sample (curve 1), the SED sample (curve 2), and original extract EXT (curve 3), taken in equal amounts (100 μL). It can be seen that all of them exhibited the ability to quench chemiluminescence, with the initial extract being the most active, while the supernatant was the least active. A comparison of the activities of these samples is shown in Figure 2B. It can be seen that the most active was the initial extract (curve 1), then the sample containing the wash from the filter (curve 2), and only then the protein-free supernatant (curve 3). Interestingly, the protein-free supernatant, which had an insignificant dry matter content, still exhibited a very noticeable ability to inhibit the development of chemiluminescence, corresponding to the presence of nonprotein antioxidant substances in the extract.

The main result of determining the AOA activity of the original beetle extract is shown in Figure 2C, which indicates the dependence of the decrease in the amplitude of chemiluminescence on the concentration of dry substances in the original extract. Comparison of the AOA value with that for the water-soluble vitamin E analog of Trolox (Figure 2D) showed that the beetle extract at a concentration of 1 mg/mL (1 g/L) exhibited the same inhibitory effect as 0.2 mM Trolox.

The *U. dermestoides* extract influenced the median lifespan (MLS) of the nematodes *C. elegans*, both under standard conditions and under paraquat-induced oxidative stress. In the presence of an undiluted extract (10% of the total sample volume), the MLS of nematodes in the absence of stress increased by 34% (predicted value—47 h in the experiment versus 35 h in the control). When the extract was diluted four times (corresponding to 2.5% of the undiluted extract in the total sample volume), the nematode MLS increased by 14% relative to the control (40 h in the experiment versus 35 h in the control). A further decrease in the dose of the extract to 1% did not have a noticeable effect—the MLS was at the level of the control group (Figure 3).

Changes in the MLS of nematodes in the presence of various concentrations of an aqueous extract of *U. dermestoides* under conditions of paraquat-induced oxidative stress are shown in Figure 4.

Under the conditions of paraquat-induced oxidative stress, the aqueous extract exhibited an AO effect at a concentration of 10%, significantly prolonging the nematode MLS by 17%; in contrast, the extract diluted 10 times did not have a noticeable effect on the MLS (Table 3).

The inhibitory effect of the beetle extract on the fructosylation of bovine serum albumin is shown in Figure 5, which indicates the kinetics of the increase in the fluorescent products of BSA modification in the control (no extract) and in the presence of the beetle extract (plus extract).

As can be seen from Figure 5, the beetle extract at a concentration of 360 μg/mL led to a pronounced slowdown in the accumulation of fluorescent products in albumin.

## 4. Discussion

Natural antioxidants can be essential for solving health problems, lifespan, and improving quality of life. However, insect antioxidants are much less studied than plant antioxidants. Some insect species contain very high concentrations of antioxidants, even when compared to classic herbal products such as olive oil or orange juice. For example, it has been shown that Trolox equivalent antioxidant capacity (TEAC) of extracts of some edible insects (grasshopper, silkworm, cricket) is five times higher than the TEAC of fresh orange juice. Simultaneously, the extracts of grasshopper and cricket exhibited high Ferric reducing antioxidant power, twice as high as that for freshly orange juice [8]. It is also known for the high antioxidant and anti-inflammatory activity of hydrolysates of edible insects—cricket *Gryllodes sigillatus*, mealworm *Tenebrio molitor*, and locust *Schistocerca gregaria* [6].

In this work, we have found that the aqueous extract of the *U. dermestoides* beetle obtained by the EPDE method is a strong quencher of the chemiluminescence of luminol, which indicates the content of antioxidants in the extract. The high antioxidant activity of the aqueous extract of the *U. dermestoides* beetle is due to the protein and nonprotein antioxidants contained in it. The antioxidant proteins included superoxide dismutase (SOD), one of the most important enzymes of the body’s antioxidant system. Heat shock proteins with molecular weights of 60, 70, and 83 kDa [31] likely play an important role in the biological activity of the extract under conditions of oxidative stress, protecting biological systems from damaging effects under stress, including oxidative stress. The presence of a poorly studied vitellogenin-like protein in an amount exceeding the SOD content was also noted. Vitellogenin is an insect antioxidant that may affect longevity [32]. According to the literature, studies of the molecular mechanisms of Vg function in worker bees have shown that Vg protects tissues from oxidative damage, which usually occurs during aging in cells or tissues of various species [33].

The aqueous extract also contains nonprotein substances with anti-infective properties, such as phenolic compounds and ethyl p-hydroquinone, as well as methyl esters of widespread octadecenoic (oleic) and hexadecanoic fatty acids. Octadecenoic acid is a monounsaturated fatty acid that refers to the group of omega-9 unsaturated fatty acids and is one of the most important acids contained in lipids involved in the construction of biological membranes.

The aqueous extract of the beetle showed an inhibitory effect on the Maillard reaction. The Maillard reaction (or nonenzymatic glycation) is a complex reaction process between the aldehyde (or ketone) groups of reducing sugars and the amino groups of proteins, as a result of which—under conditions of hyperglycemia—the so-called advanced glycation end products (AGEs) are formed. It is known that AGE products of glycation, such as advanced lipoxidation end products (ALEs), are extremely toxic to cells [34]. The formation and accumulation of AGE products in various cells and tissues leads to damage to extracellular and intracellular structures and disruption of their functions and is one of the main reasons for the development of diabetic complications [35]. The accumulation of AGE products occurs in aging, diabetes, arthritis, atherosclerosis, chronic renal failure, nephropathy, neuropathy, Alzheimer’s disease, etc. [34,36]. The ability of beetle extracts to suppress the process of nonenzymatic glycation may be associated with both the presence of antioxidants in their composition and amine-containing substances that can compete with the amino groups of proteins in reaction with carbonyls. It is well known that antioxidants can inhibit the development of the Maillard reaction [37]. This discovered property of the considered aqueous beetle extract to inhibit the glycation process may be used in pharmacological practice, in the treatment and prevention of diseases associated with aging and diabetes.

The discovered components of the *U. dermestoides* extract have multiple biological activities, the totality of which also manifested through experiments with the free-living soil nematode *C. elegans*. The water extract of the beetle had a dose-dependent effect on the lifespan of the nematode, both under normal conditions (i.e., in the control without oxidative stress) and under OS conditions. In the presence of 10% extract, nematodes looked healthy and remained active much longer than nematodes from other groups: the MLS in the control without OS was 36 h, while, in the presence of the extract, the MLS was more than 48 h (minimum predicted value). Oxidative stress reduced the MLS of control nematodes up to 24 h. However, with the simultaneous introduction of 10% of the extract, the MLS increased up to 28 h (more than 16%). Dilution of the extract (reducing the injection dose to 2.5%) reduced the effect of the extract on the MLS, where no biological activity was observed at a dose of 1%. It can be assumed that the presence of antioxidants in the *U. dermestoides* extract was associated with the effect of increasing the lifespan of the nematode *C. elegans*, both under normal conditions and under oxidative stress.

## 5. Conclusions

We derived an aqueous extract from the *U. dermestoides* beetle using the electro-pulse plasma dynamic extraction method and demonstrated that it features a complex of antioxidants that show promise for further research on their biological activity. At the same time, the presence of relatively large amounts of phenol-containing substances, such as resorcinol, indicates a possible toxic effect; therefore, the practical use of this extract requires caution and study of its toxicity.

## Figures and Tables

**Figure 1 antioxidants-10-01055-f001:**
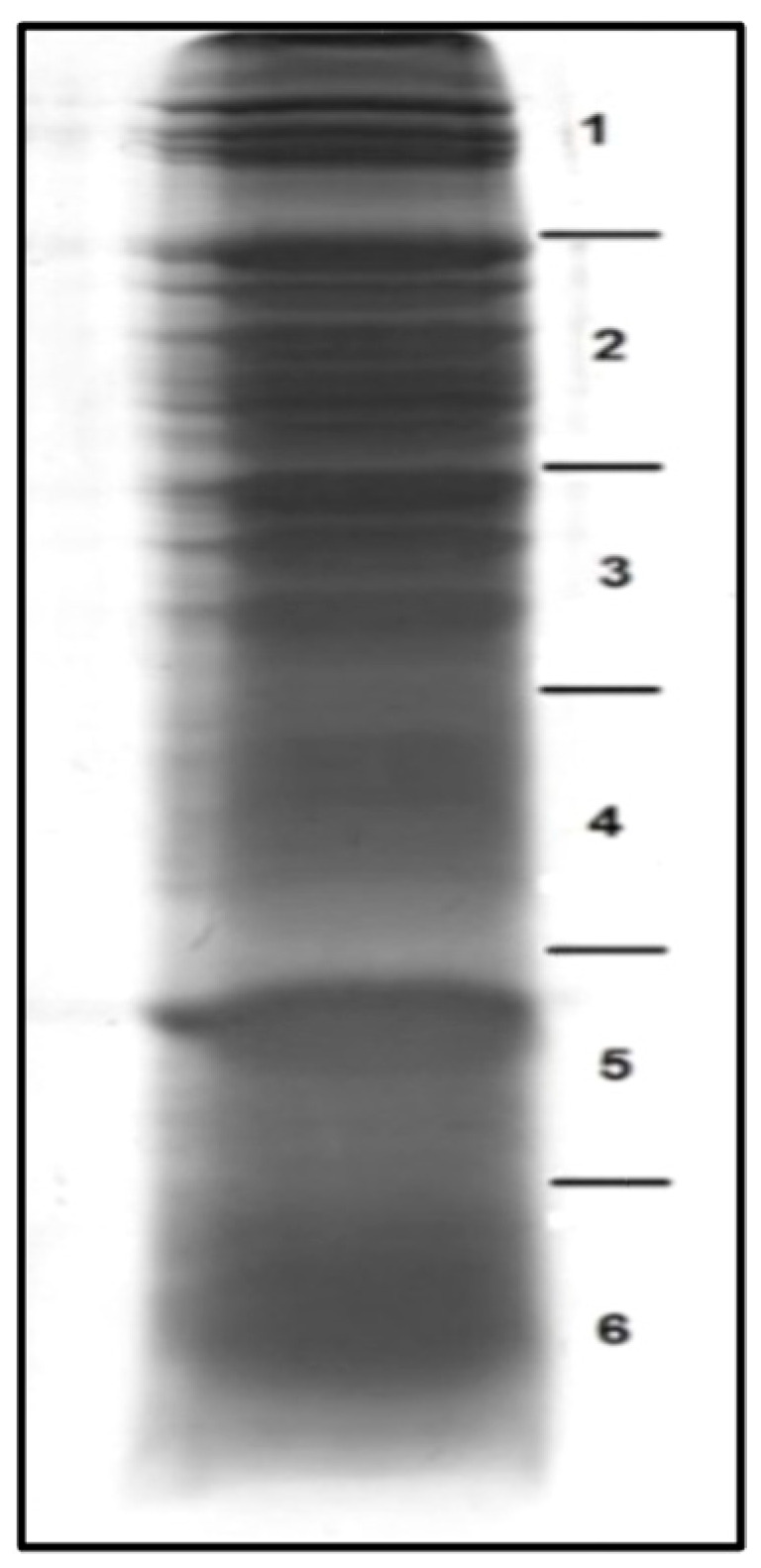
Image of protein separation by SDS–PAGE with scheme of cutting for six gel fragments.

**Figure 2 antioxidants-10-01055-f002:**
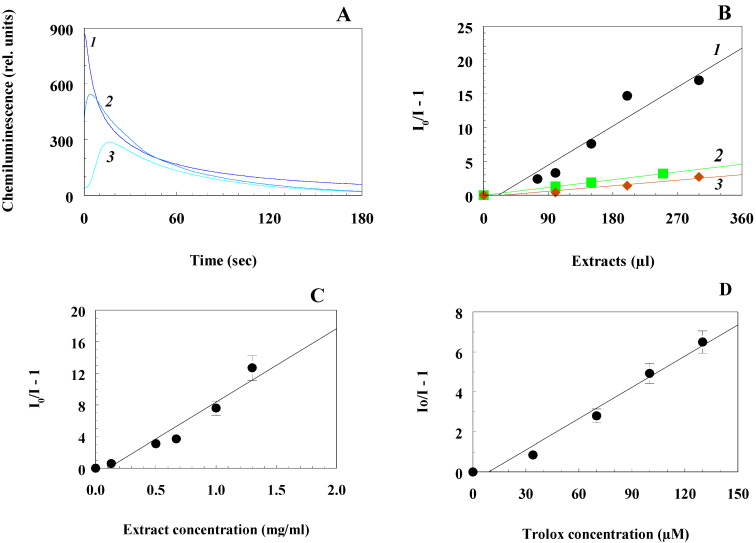
Determination of the antioxidant activity of the aqueous extract of the beetle by the chemiluminescent method: (**A**) quenching of luminol chemiluminescence by the original extract of the beetle (curve 3), the supernatant from it with the removed proteins (curve 1), and the remaining proteins (curve 2). All additives are 100 μL of samples; (**B**) comparison of AOA for the original extract from the beetle (curve 1), protein-free supernatant (curve 3), and the filter residue (curve 2); (**C**) determination of the value of AOA of the whole (original) extract of the beetle; (**D**) determination of the value AOA of Trolox.

**Figure 3 antioxidants-10-01055-f003:**
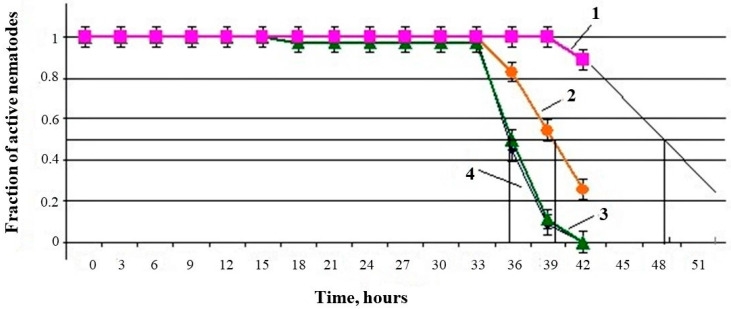
Changes in the median lifespan of nematodes with time in the presence of different concentrations of *U. dermestoides* extract: (1, predicted value) 10% of the total sample volume; (2) 2.5%; (3) 1%; (4) control.

**Figure 4 antioxidants-10-01055-f004:**
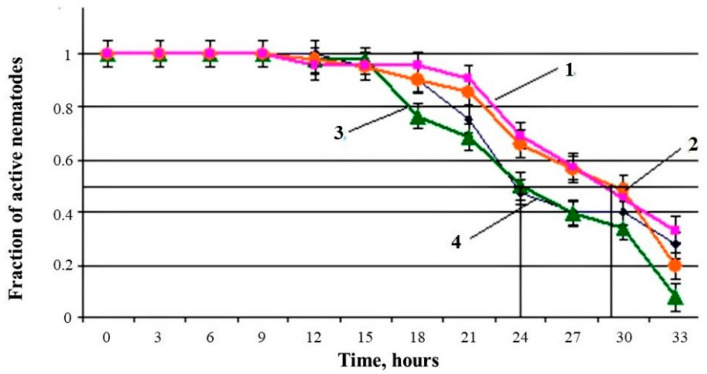
Changes in the median lifespan of nematodes with time under conditions of paraquat-induced oxidative stress in the presence of different concentrations of an aqueous extract of the beetle *U. dermestoides*: (1) 10% of total sample volume; (2) 2.5%; (3) 1%; (4) control.

**Figure 5 antioxidants-10-01055-f005:**
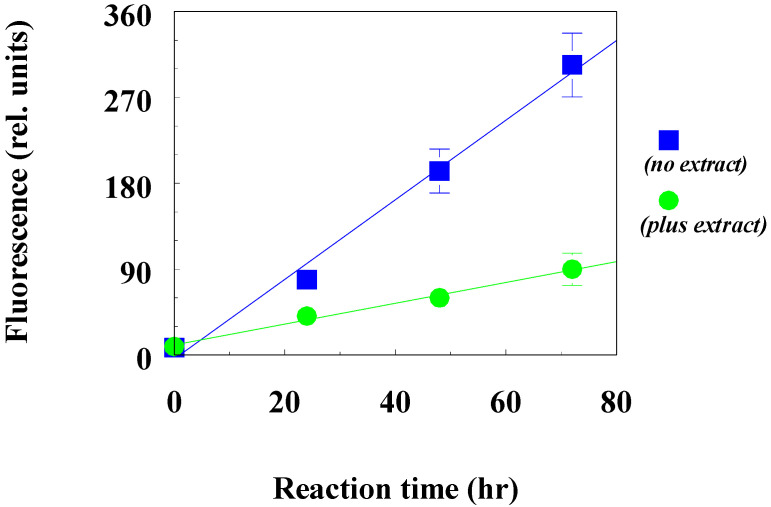
Influence of beetle extract on the rate of fructosylation of bovine serum albumin: no extract, control; plus extract, added 0.36 mg/mL dry weight of beetle extract. Incubation was carried out at a temperature of 37 °C in the dark and with constant stirring.

**Table 1 antioxidants-10-01055-t001:** Proteins identified by homology—antioxidants and heat shock proteins in an aqueous extract of the *U. dermestoides* beetle.

No.	Name of Protein (Fasta Headers)	Mw, kDa	Protein Content in the Extract, %
Antioxidant proteins			
1	tr|D2A2T2|D2A2T2_TRICA Superoxide dismutase [Cu-Zn]	15.689	1.13
2	tr|A0A139WLA9|A0A139WLA9_TRICA Catalase	54.648	0.61
3	tr|Q49I38|Q49I38_TRICA Pro-phenol oxidase subunit 2	79.369	0.26
4	tr|D6W9X8|D6W9X8_TRICA Peroxiredoxin 1-like Protein	27.498	0.09
5	tr|Q49I39|Q49I39_TRICA Phenoloxidase subunit A3-like Protein	79.1	0.08
6	tr|D6WDX1|D6WDX1_TRICA Thioredoxin reductase 1, mitochondrial-like Protein	53.58	0.04
7	tr|D2A3X3|D2A3X3_TRICA Peroxiredoxin 1-like Protein	21.778	0.01
8	tr|D6W721|D6W721_TRICA Vitellogenin-like Protein	204.05	1.21
Heat shock proteins			
1	tr|D6WD04|D6WD04_TRICA Heat shock 70 kDa protein cognate 3-like Protein	72.685	0.64
2	tr|D6WKD1|D6WKD1_TRICA 60 kDa heat shock protein, mitochondrial-like Protein	61.131	0.39
3	tr|A0A139WAS5|A0A139WAS5_TRICA Heat shock 70 kDa protein cognate 4-like Protein	71.08	0.23
4	tr|D6WA11|D6WA11_TRICA Heat shock 70 kDa protein cognate 5-like Protein	75.34	0.04
5	tr|D6WCB2|D6WCB2_TRICA Heat shock 70 kDa protein cognate 2-like Protein	69.132	0.02
6	tr|D6X0J9|D6X0J9_TRICA Heat shock protein 83	89.532	0.01
7	tr|D6WU59|D6WU59_TRICA Heat shock protein 68a	70.866	0.001

**Table 2 antioxidants-10-01055-t002:** Composition of nonprotein components of the aqueous extract of the *U. dermestoides* beetle.

RT, min	RI Calculation	m/z	Compound	Content, μg/g
		41–140	The amount of extractable compounds	54.6 ± 5.4
11.47	1084	66;94; 124	3-methoxy phenol;	0.2 ± 0.1
13.5	1200	79;108;123;138	3-methoxy-2-methylphenol *	5.2 ± 1.0
17.55	1429	95;124	Methylresorcinol *	0.4 ± 0.1
19.13	1519	123;138	Ethyl p-hydroquinone	11.5 ± 1.7
20.08	1573	69;97	1-tridecanol *	2.1 ± 0.4
21.84	1673	121;138;166	Ethyl p-ethoxybenzoate	0.5 ± 0.1
27.46	1993	74	Methyl ether of hexadecanoic acid	2.0 ± 0.4
29.95	2134	264	Octadecenoic acid methyl ester	4.3 ± 0.9
30.26	2152	74	Methyl ether of octadecanoic acid	0.3 ± 0.1

* Identification is tentative.

**Table 3 antioxidants-10-01055-t003:** MLS of the nematode *C. elegans*.

Test Extract	Concentration of Test Extract in the Cell, %	Number of Live Nematodes per 35 h M ± SE under Normal Conditions (without PQ)	MLS under Normal Conditions (without PQ)		Number of Live Nematodes per 24 h M ± SE with OS (with 50 mM PQ)	MLS with OS (with 50 mM PQ)	
			M ± SE Hour	%		M ± SE Hour	%
10% Lightning extract	10.0	32.5 ± 4.95 *	47 ± 0.51	134	28 ± 1.4 *	29 ± 0.31	117
Control	0.0	17 ± 1.41	35 ± 0.38	100	19 ± 0.71	24 ± 0.25	100

* The difference between the option with the extract–lightning concentrate and the control is statistically significant (F = 85.03; *p* ≤ 0.01).

## Data Availability

Data is contained within the article.

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
