# Peer review of "Novel Extract from Beetle Ulomoides dermestoides: A Study of Composition and Antioxidant Activity"

_antioxidants, 2021, doi:10.3390/antiox10071055_

Round 1

Reviewer 1 Report

Introduction: In my opinion emphasis should be given on relevant studies and the extraction method used, which is a novel one, and discuss around this, rather than the methods used for analysis (Lines 50-71). Analytical methods should be mentioned in brief in this section following the scope of the study.

Antioxidant activity: Typically, the antioxidant activity of an extract is determined with more than one method. For instance, DPPH is often used along with FRAP etc to compare results. DPPH scavenging represents a non-biological system, whereas FRAP a biological one. This also allows comparison with other studies. Maybe the authors could justify the use of chemiluminescence as the sole method for AOA.  

Discussion section: discussion should be improved and findings should be compared to other relevant studies on insect extracts.

Lines 349-352: In my opinion Trolox data on the inhibitory effect should also be included in Figure 2C as it is the reference compound.

Minor corrections:

Line 33-Reference required for this statement.

Line 52- replace “experiments” with “systems”

Line 53- replace “for” with “to”

Line 54- replace “to” with “with”

Introduction: Lines 73-75 before to 66-72

Line 90: what is the full name for EIPDE method? Is it same with EPDE?

Lines 92-94: use past tenses instead of present.

Line 100: Report the authors of citation 14.

Lines 101-103: “in accordance with standard techniques, manipulations with nematodes were carried out according to standard techniques”. Improve the the language use in this sentence.

Lines 254-256: Include a reference for the expression of antioxidant activity with the MLS.

Line 264: Specify the test used for statistical analysis.

Line 384: “shown in Fig. 5, which shows…”, use another verb instead of show twice in the sentence.

Lines 413-423: Appropriate references should be included for these statements.

Author Response

Dear Reviewer,

Thank you for fast consideration of our manuscript (antioxidants-1252993). I studied your remarks. It seems to me remarks quite reasonable. I would like to thank you for careful studying of our manuscript and for critical remarks. It always is useful for improvement quality of work. We made all (hopefully) the necessary changes to the text of the manuscript.
We've changed the Introduction to include information about the extraction EPDE method. Included all required missing references (lines: 33; 254-256; 413-423 reviews). Literature data on the antioxidant activity of insect extracts were added to the Discussion section. The authors of the manuscript agree with your opinion that the results of the inhibitory effect of Trolox should be included in Figure 2. We have added this data in the form of Figure 2D.
Now for a comment on the methods for determining the antioxidant activity of the extract. The study of the antioxidant activity of both individual substances and biological fluids by the method of quenching chemiluminescence has long been successfully used at our Institute. The method can be used to assess the total antioxidant activity in various biological tissues, for example, in saliva, in serum, in tea, etc., see, for example, Hirayama O., Takagi M., Hukumoto K., Katoh S. Evaluation of antioxidant activity by chemiluminescence. Anal. Biochem., 1997, 247(2), 237-241. Doi: 10.1006/abio.1997.2053. Of course, it is interesting to compare the antioxidant activity of one object with the methods of quenching chemiluminescence, interaction with stable radicals DPPH and with the method where the reaction Fe3+-TPTZ → Fe2+-TPTZ is used, but this is a subject for future research.

Sincerely yours,
Dr. Alexander Dontsov

Reviewer 2 Report

The reviewed manuscript Novel Extract from Beetle Ulomoides dermestoides: A Study of Composition and Antioxidant Activity is correctly written. The aim of the study was to analyze the composition and assessment of the biological activity of the water extract of biomass of adult sexually mature U. dermestoides. The research was planned and approved by the Ethics Committee of A.N. Severtsov Institute of Ecology and Evolu-110tion of Russian Academy of Sciences (Protocol Code: 009; Date of approval: 10/26/2020).

Reviewer's comments:

Line 264: The results were well described, although statistical research needs to be completed.

Please describe what were performed using Statistica 12.

Line 246: should be 20oC

Lines 353-361 and Figure 3. How did the authors determine the end point for curve 1. One should be very careful with the extrapolated values. Forecasting variable time based on one variable can be subject to significant error. Please note that only one point for 10% is below the value of 1 active nematodes. I suggest you make it clear that this is the predicted value.

References. About 50% of the citations are more than 10 years old. Have the authors found newer literature?

In tables with numerical data, commas should be replaced with dots and standard deviations should be given.

In my opinion, the manuscript should be edited as suggested. 

Author Response

Dear Reviewer,

Thank you for fast consideration of our manuscript (antioxidants-1252993). I studied your remarks. It seems to me remarks quite reasonable. I would like to thank you for careful studying of our manuscript and for critical remarks. It always is useful for improvement quality of work.
We made all the necessary changes to the text of the manuscript.
We completely agree with you that the end point of curve 1, Fig. 3 – the result of extrapolation and has only a predicted value. We added this to the text.
In tables with numerical values, commas were replaced with dots. We make changes into the tables, wherever possible. Unfortunately, we were unable to enter statistical data into Table 1. This is due to the following reasons. Label-free protein quantification is based on iBAQ provided by MaxQuant software report. This iBAQ value results from the sum of individual protein`s iBAQs for each technical repeat of each gel fragment being analyzed. The percentage of a specific protein ("Protein content in the extract, %" column) was calculated as the ratio of the iBAQ value of the protein to the sum of all iBAQ values for all identified proteins, multiplied by 100%. We have no possibility to calculate standard deviation because we have only one summarized iBAQ value per protein. The main goal of proteomic analysis of the beetle extract in this study is a qualitative evaluation of the protein composition.
We also reduced the relative number of citations over 10 years old by adding additional links. However, I believe that replacing all cited works with more young ones is not entirely fair for their authors. Sometimes it is simply impossible to do it.

Sincerely yours,
Dr. Alexander Dontsov

Round 2

Reviewer 1 Report

The manuscript has been properly revised, taking into account all proposed recommendations.

For the antioxidant activity of course methods vary according to available equipment and suggested methods like dpph are helpfull for future work.

I have no further remarks on the manuscript.

Reviewer 2 Report

The Authors revised the manuscript in line with the previously submitted comments.